# Evaluation of Smallpox Vaccination Coverage and Attitude towards Monkeypox Vaccination among Healthcare Workers in an Italian University Hospital

**DOI:** 10.3390/vaccines11121741

**Published:** 2023-11-22

**Authors:** Sergio Scarinci, Martina Padovan, Bianca Cosci, Armando Petillo, Vittorio Gattini, Francesca Cosentino, Aldo Mignani, Rudy Foddis, Giovanni Guglielmi

**Affiliations:** 1Occupational Preventive Medicine Ward, Azienda Ospedaliero-Universitaria Pisana, 56124 Pisa, Italy; scasers@gmail.com (S.S.); padovan.martina@gmail.com (M.P.); coscibianca@gmail.com (B.C.); armando.petillo@virgilio.it (A.P.); v.gattini@ao-pisa.toscana.it (V.G.); f.cosentino@ao-pisa.toscana.it (F.C.); a.mignani@ao-pisa.toscana.it (A.M.); 2Department of Translational Research and New Technologies in Medicine and Surgery, University of Pisa, 56126 Pisa, Italy; rudy.foddis@unipi.it

**Keywords:** vaccine coverage, monkeypox, Mpox, smallpox, healthcare workers, occupational health, occupational exposure, vaccination attitude

## Abstract

(1) Background: In 2022, monkeypox (Mpox) was declared a public health emergency. The European Medicines Agency has authorized the use of Imvanex/Jynneos, a smallpox vaccine, for coverage against pox. Healthcare workers (HCWs) are all considered by the European Centre for Disease Prevention and Control to be at risk, but in Italy, vaccination was offered only to laboratory personnel. The present study aims to investigate smallpox vaccination coverage (VC) that provides protection against Mpox among HCWs in an Italian university hospital and to assess HCWs’ attitudes towards the possibility of getting vaccinated against Mpox. (2) Methods: We conducted a cross-sectional survey. 336 HCWs from selected wards were asked to fill out a self-declaration to collect their sex, profession, ward, vaccination status, and attitude toward Mpox vaccination. (3) Results: 60.71% of HCWs involved provided the requested data; 38.7% of them were previously vaccinated against smallpox, which corresponds to 23.5% of the total HCWs in the wards considered. Considering those born before 1979 as vaccinated, VC increases from 23.5% to 41.7%; the percentage of HCWs who adhered to vaccination is 23%; laboratory technicians showed a lower willingness to be vaccinated. The ward with the highest willingness to vaccinate is proctological surgery. (4) Conclusions: Based on our experience, a variability in smallpox VC and in willingness to vaccination has emerged both among different job titles and age categories and across the wards analyzed. Additionally, our survey reveals that vaccination attitudes are higher among HCWs from wards that currently do not have free access to such vaccinations.

## 1. Introduction

### 1.1. Monkeypox

Mpox (formerly named monkeypox) is a viral infection known as human monkeypox (HMPX) and is caused by the monkeypox virus (MPV), a type of DNA virus member of the orthopoxvirus, which is part of the Chordopoxvirinae subfamily and the Poxviridae family [1].

After the eradication of smallpox in 1980 and the discontinuation of universal smallpox vaccination, Mpox has emerged as the predominant zoonotic disease caused by orthopoxvirus infection in humans [2].

On 23 July 2022, the World Health Organization (WHO) declared the global Mpox outbreak a public health emergency [3].

Since the start of the Mpox outbreak and as of 7 June 2023, a total of 25,910 cases of Mpox have been reported from European countries [4].

In Italy, 957 cases have been confirmed up to 07 June 2023 (no other cases have been recorded since that date) [5].

In this outbreak, most reported cases have been transmitted through sexual contact with multiple partners, but Mpox can also spread via close physical contact with skin rashes, crusts, bodily fluids, and respiratory secretions during medical assistance [6,7]. 

In the context of the Mpox multi-country outbreak, mass vaccination is not recommended, but targeting primary (pre-exposure) preventive vaccination (PPV) to population groups at highest risk may help outbreak control. According to the ECDC, high-risk groups are individuals who identify themselves as homosexual, bisexual, or other men or transgender people who engage in sexual activity with men, and specific categories of workers, like healthcare workers (HCWs) [8].

The European Centre for Disease Prevention and Control (ECDC) evaluates the level of risk for HCWs in different scenarios. When wearing suitable personal protective equipment (PPE) and applying the correct precautions, the risk is considered low. However, the risk increases to moderate in situations where HCWs are exposed to a Mpox case without appropriate PPE for an extended period of time, perform procedures that generate aerosols without PPE, or experience occupational exposure in a laboratory lacking proper PPE or equipment. When HCWs use appropriate PPE and correct precautions, the likelihood of exposure is very low, resulting in an overall low risk of contracting Mpox. For laboratory personnel, the risk assessment remains consistent with the initial evaluation and depends on their use of proper PPE and adherence to appropriate laboratory protocols. Furthermore, the risk may be heightened for HCWs and laboratory personnel who are older or have weakened immune systems, as they are more susceptible to experiencing significant effects from exposure [9].

ECDC then gives an indication to consider vaccination for occupational exposure of HCWs, especially those at repeated risk of exposure, laboratory personnel, and outbreak response staff based on risk assessment [10]. 

Since July 2022, the European Medicines Agency (EMA) has authorized the use of Imvanex/Jynneos, a third-generation non-replicating smallpox vaccine, for protection against Mpox in adults [11]. 

The recommended vaccination course consists of two shots, with the second one administered at least 28 days after the first one. This two-shot regimen is intended as the primary vaccination for individuals who have not been previously vaccinated against smallpox, Mpox, or vaccinia viruses. Individuals who have received prior vaccinations against smallpox, Mpox, or vaccinia viruses may consider a single booster vaccination shot [12].

### 1.2. The Italian Context

In Italy, vaccination strategies are mostly based on primary preventive vaccination (PPV) rather than post-exposure prophylaxis. Considering the current epidemic scenario and the limited availability of vaccine shots, in August 2022, the Ministry of Health instructed that smallpox vaccination in the healthcare sector would have been initially offered to laboratory personnel with possible direct exposure to orthopoxvirus [13]. The vaccination regime followed in Italy is the one recommended by the EMA. For anyone who has received at least one shot of the smallpox vaccine or Imvanex/Jynneos (MVA-BN) in the past or who has completed the two-dose vaccination course of Imvanex/Jynneos more than two years ago, only one shot (booster vaccination) is scheduled. People who have been vaccinated against smallpox in the past may have some protection against Mpox. However, it is unlikely that people younger than 40–50 years of age have been vaccinated, since smallpox vaccination in Italy was suspended in 1979 and officially repealed in 1981 [14]. Evidence of a previous smallpox vaccination can usually be found as a scar on the upper arm on the deltoid muscle. 

Furthermore, the Italian Ministry of Health, after initial guidance (August 2022) on categories at risk, such as laboratory personnel, has not issued further guidance to extend vaccine offers to HCWs engaged in frontline work and presumably exposed to possible Mpox infection. Instead, the ECDC has given guidance on evaluating the vaccination of HCWs based on risk assessment.

In accordance with Italian law [15], vaccination prophylaxis for HCWs must be guaranteed for those who are classified as at higher risk of infection. The administration of vaccines is under the responsibility of the physician in charge of implementing the health surveillance program (“competent physician”).

On 11 May 2023, the WHO Director-General declared that Mpox is no longer a global health emergency, but the International Health Regulations (IHR) Emergency Committee also emphasized the necessity to continue to make vaccines available for primary preventive (pre-exposure) and post-exposure vaccination for people and communities at high risk of Mpox [16]. 

However, ECDC has issued a cautionary notice regarding the potential rise in cases expected during the upcoming summer season (2023). This increase could be associated with various celebratory gatherings, like Pride events, and the usual increased travel during the summer. The ECDC emphasizes the importance of the implementation of adequate vaccination strategies and health promotion in response to the increased infection incidence as vital steps for the successful control of the outbreak [17]. 

### 1.3. Aim of the Study

Data on smallpox vaccination coverage (VC) in HCWs are limited in the international literature [18], and so far, few research studies addressing this matter have been conducted in Italy [19,20,21]. Therefore, it is crucial to gather more data to determine effective strategies for enhancing VC to protect HCWs from Mpox.

The present study aims to investigate the current smallpox VC within some medical wards classified as low, moderate, and high risk for Mpox infection in an Italian university hospital. Secondly, we assessed the HCWs’ attitude towards the Mpox vaccination.

This study provides evidence on the degree of vaccination coverage and vaccination attitude in a university hospital, potentially useful for improving future vaccination strategies for HCWs during infectious epidemics.

## 2. Materials and Methods

### 2.1. Setting

The Azienda Ospedaliero-Universitaria Pisana (AOUP) is the most important hospital in the north-west area of Tuscany. The AOUP encompasses 10 divisions and 158 structures, employing more than 5000 individuals. This center has a medium-large volume of activity. In 2019, it had 94,436 Emergency ward admissions, 57,090 total hospitalizations and 296,532 outpatient specialist consultations. 

Out of the 957 reported cases of Mpox in Italy as of 4 July 2023, 47 were observed in Tuscany, of which 12 were identified and reported in AOUP (all between 28 July 2022 and 26 August 2022), and no other cases have been registered in AOUP since then.

Since the end of September and the beginning of November 2022, in AOUP, three HCWs from the Virology ward have been vaccinated against Mpox with two shots (separated by 28 days) of the Jynneos vaccine.

### 2.2. Study Population

The study population consists of HCWs working in AOUP.

### 2.3. Study Sample

HCWs working in AOUP wards considered both at high and low/moderate risk for exposure to Mpox have been included in the study. The risk assessment is explained in the Section 2.7. All job titles not involving either exposure to infectious patients or manipulating biological specimens were excluded.

After the application of exclusion criteria, the studied subjects were 336 (10 were excluded for the administrative job profile). In our sample, the mean age is 43.2, with a standard deviation of 10.6. There are 236 females and 100 males. The overall response rate was 60.71%. The sample size was determined empirically based on the design of similar studies [20,22]. The probability value for statistical significance was determined to be 0.05.

### 2.4. Study Design

We conducted a cross-sectional survey between June 2022 and April 2023 involving HCWs from AOUP wards considered both at high risk and at low/moderate risk for exposure to Mpox.

### 2.5. Self-Declaration Development

The self-declaration was part of the routine procedure used during the periodical health surveillance program with the aim of collecting consensus for vaccination. The self-declaration collected information about the socio-demographic characteristics of participants (i.e., gender, age, job title, and type of ward), smallpox vaccination status, and the intention to get the vaccine against Mpox. The form was to be filled out by the HCWs certifying their previous VC for smallpox (checking their childhood vaccination certificate for a double smallpox shot or checking the presence of vaccination scars on their upper arm) and reporting adherence to the smallpox vaccination to protect against Mpox, with a complete cycle or with just a booster shot in case of a vaccination offer. 

### 2.6. Data Collection

According to the above-mentioned procedure, between June 2022 and April 2023, the self-declaration was sent via email using the internal mailbox to the heads of the wards and nursing coordinators of the wards considered.

To maximize participation, the physician in charge of the vaccination clinic of the Occupational Medicine ward directly contacted the heads of the ward, emphasizing the importance of a complete response by as many HCWs as possible. Therefore, it was requested to print and distribute the self-declaration to all HCWs in the ward. Once completed, they were collected and sent via email to the occupational health clinic’s vaccination unit coordinator.

The data were not collected in an anonymous way since it was part of the health surveillance program. For the same reason, Ethics Committee approval was not needed.

As far as the socio-demographic information of the personnel who did not respond to the survey, we extrapolated the lacking data from the electronic personal health record implemented during health surveillance periodical exams at the Occupational Medicine ward of AOUP.

The self-declaration, translated into English, is available in the Appendix A.

### 2.7. Risk Assessment 

The risk classification of the wards included in this study simply corresponds to the results of AOUP’s risk assessment (which in Italy is a mandatory act by law), as available in the risk assessment documentation.

The table with an extract of the risk assessment documentation is shown in the Appendix A.

### 2.8. Statistical Analysis

The statistical analysis was performed using Prism v. 9. Categorical variables were expressed as frequencies, whereas quantitative variables were expressed as means or medians. The statistical differences between the groups were assessed using the Chi-Square test for categorical variables and the Mann-Whitney test or unpaired Student *t*-test for quantitative variables, as appropriate. 

The odds ratios were assessed using logistic regression. A comparison was performed by assuming females, Infectious Disease wards, and physicians as the reference groups.

The statistical significance was accepted as *p* < 0.05.

## 3. Results

### 3.1. Answers to Self-Declaration

204 enrolled HCWs who provided the requested information had a median age of 44 years (IQR 19), and 28.4% (58) were male. Response rate was uneven in the wards involved, with a much higher rate from the Virology (88.4%) and Infectious Disease wards (82.7%) compared to the Proctological Surgery (63.6%) and Emergency (50%) wards, though statistical significance was reached only with the last one (*p* < 0.001; OR: 0.209; 95% CI: 0.098–0.449).

After stratification of the HCWs by job title, nurses and laboratory technicians’ response rate was significantly higher (*p* = 0.018; OR: 1.976; 95% CI: 1.123–3.480 and *p* < 0.001; OR: 43.871; 95% CI: 5.687–338.450, respectively).

The complete results for self-report completion are shown in Table 1.

### 3.2. Vaccination Coverage

The percentage of staff previously vaccinated for smallpox was 38.7% (79 HCWs) of all respondents, which corresponds to 23.5% of the total number of HCWs in the included wards. The median age of those vaccinated was 54 (IQR 7), 59 (40.4%) were females, and 20 (34.5%) were males.

The rate of vaccination calculated for job title and ward affiliation among the recruited HCWs was as follows: 32.7% (32) of nurses; 44.4% (12) of physicians; 62.9% (22) of healthcare assistants; 33.3% (1) of midwives; 35.3% (12) of laboratory technicians; 60.5% (26) of Infectious Diseases; 32.8% (38) of Emergency Medicine; 31.6% (12) of Virology; and 42.9% (3) of Proctological Surgery ward, with some of these differences resulting in statistically different results (Table 2). As expected, the difference in the age median between vaccinated (54 IQR7) and unvaccinated (37 IQR19) was different (*p* < 0.001; OR: 1.51; 95% CI: 1.330–1.710). 

The Emergency Medicine ward median age of the HCWs is 41 (IQR 15.75) and the Virology ward median age is 44 (IQR 22); these values are lower compared to the median of HCWs from both the Proctological Surgery and Infectious Diseases wards (50 IQR 19; *p* < 0.001; *p* = 0.025).

More details with regard to the VC are shown in Table 2.

Analyzing the 132 HCWs who did not respond to the self-declaration, 61 HCWs were born before 1979 (the year when the vaccination requirement for smallpox was repealed in Italy). If we consider those born before 1979 as vaccinated, the VC increases from 23.5% (79 HCWs) to 41.7% (140 HCWs), of which 33 (23.6%) belong to the Infectious Diseases ward, 85 (60.7%) to the Emergency Medicine ward, 17 (12.1%) to the Virology ward, and 5 (3.6%) to the Proctological Surgery ward.

Data on the stratification by year of birth (born before 1979 vs. born since 1979) and ward of HCWs that did not fill out the self-declaration are shown in Appendix A, reported in Appendix A.

### 3.3. Willingness to Receive Mpox Vaccination

In terms of willingness to adhere to vaccination, the percentage of staff who declared their willingness to be vaccinated was 23% (47 HCWs). Out of these 47 (24.1% male and 22.6% female), 11 (23.4%) were from Infectious Diseases, 25 (53.2%) from Emergency Medicine, 5 (10.6%) from Virology, and 6 (12.8%) from the Proctological Surgery ward.

In terms of job title, 1 biologist (2.1%), 21 (44.7%) nurses, 11 (23.4%) physicians, 8 (17%) healthcare assistants, 3 midwives (6.4%), and 3 (6.4%) laboratory technicians adhered to the vaccination. Among the 47 HCWs receiving the Mpox vaccines, 10 (21%) required only one booster shot as they had been previously vaccinated, while 37 (79%) needed full vaccination as they had never been vaccinated before. Notably, the age difference between HCWs who declared willingness to adhere to vaccination and HCWs who did not was statistically significant (*p* = 0.01; OR: 0.959; 95% CI: 0.930–0.989).

The higher willingness to be vaccinated was registered among HCWs from the Proctological Surgery ward (*p* = 0.012 OR: 17.455–95% CI: 1.886–161.528), while, as far as the job title is concerned, the laboratory technicians showed the lowest willingness rate (*p* = 0.015 OR: 0.176–95% CI: 0.044–0.710), see Table 3.

## 4. Discussion

The aim of this study was to investigate smallpox VC, which provides protection against Mpox among HCWs working in selected wards based on the risk assessment in an Italian university hospital, and to assess HCWs’ attitudes towards the vaccination against Mpox.

Regarding the self-declaration analysis, the overall response rate (60.71%) was similar to previous studies with comparable sample sizes [20] and higher than studies with a larger sample size [19].

When considering the answer stratified by job title, significant differences emerged only among nurses and laboratory technicians. Our data agree with the findings in the comprehensive literature, which typically indicates that nurses exhibit higher responsiveness to surveys [22,23,24].

Regarding the VC, the Virology and Emergency Medicine ward has a lower number of previously vaccinated HCWs. This data can be explained if we consider the average age of HCWs in the respective wards in correlation with the year in which smallpox vaccination was suspended in Italy [14].

Among the HCWs, healthcare assistants have the highest VC, possibly due to their higher average age.

The median age of those who reported being previously vaccinated is 54 years (IQR 7), while the median age of those who reported not being vaccinated is 37 years (IQR 19) (*p* < 0.001). This is easily explained by the fact that smallpox vaccination in Italy was suspended in 1979, so people 45 or younger were not vaccinated in their childhood; coherently, we did not find any difference in sex distribution among those who were previously vaccinated. 

The VC estimated in our study, including people born before 1979, was similar to that shown in another similar study [19].

Such higher VC related to age is consistent with a well-established and widespread trend among HCWs for flu vaccinations [25], and for vaccinations in general as well [26], and it is also consistent with what has been previously found in the AOUP regarding VC in HCWs for flu and meningitis [27,28]. 

Our results show a lower rate of willingness to receive smallpox vaccination in comparison with other similar studies [19,29,30]. Most likely, this data has to do with the median age of our sample population (38 years, 17.5 IQR), which is significantly younger than the median age of other studies.

Regarding adherence to the smallpox vaccination, there is no gender difference, and this finding contradicts the few studies that have assessed vaccine willingness in the literature. In a study by Harapan et al., men had lower odds of being willing to take the vaccine than women [31], while in Riad et al., the smallpox vaccine acceptance level to protect against Mpox in HCWs was higher among males than females [32], as well as in Ghazy et al., where males were more confident regarding smallpox vaccination [33]. 

When stratifying by ward, those who statistically showed the highest adherence to vaccination were the healthcare assistants in the Proctological Surgery ward, probably because they assist physicians during the clinical examinations and are assigned to take care of the patients who may present a typical skin manifestation in the anal region. 

Regarding the job title, laboratory technicians showed a lower willingness to vaccinate, despite being the only category of HCWs for whom smallpox vaccination is currently indicated in Italy. Likely, their attitude towards vaccination is due to a low-risk perception, even though this should be confirmed by a specific investigation.

Our data show that the VC is low, which suggests a need to strengthen vaccination policies.

In the past, the adoption of such a vaccination policy had already been implemented during the meningococcal outbreak that affected the Tuscany region in 2015. The vaccination offered in our hospital was extended to all HCWs. Until that point, according to the WHO, anti-meningococcal vaccination (anti-Men ACWY) was recommended only for laboratory personnel working in Microbiology wards [34], and in Italy, it was neither mandatory nor recommended for other HCWs, except for laboratory staff [35]. During this outbreak in our hospital, following the interim guidelines of the Tuscany region, except for administrative staff and personnel over the age of 75, vaccination was offered by the preventive medicine staff during the medical examination for occupational purposes. From 2015 to 2017, 41.1% of HCWs voluntarily received the meningococcal vaccination. Certainly, in this case, media attention played a significant role due to the increase of meningitis cases in the Tuscany region. However, it serves as an example of how expanding vaccination offerings to all HCWs ensures an increase in VC for that specific pathogen, even though the VC for Neisseria meningitidis among HCWs in our hospital is still suboptimal [28].

Our study has some limitations. The smallpox VC and attitude to vaccinate against Mpox were evaluated only in a limited number of AOUPs, specifically those with a low to moderate/high risk. Only 60.71% of the involved HCWs completed the self-declaration. Probably due to a poor knowledge of the Mpox disease and a low-risk perception; furthermore, we have not evaluated the level of knowledge of HCWs toward the Mpox virus (epidemiology and clinic features). Lack of investigation of Mpox knowledge is another important limit of our study that generates relevant biases like “non-responded” and “information bias”. In fact, we did not assess the risk perception of those who refused the vaccination.

Finally, our study represents the experience of a single Italian hospital. Hence, it is important to conduct a multicenter study encompassing a larger cohort of HCWs to validate our findings.

## 5. Conclusions

The incidence of Mpox cases in Italy appears to have shown a significant decline in 2023. Even though there has been a substantial reduction in Mpox cases, it is crucial to maintain high attention, especially for HCWs involved in patient care.

Based on our experience and the assessment of Mpox infection risk in AOUP, it emerges that the risk of Mpox infection is not confined solely to the Virology ward, whose laboratory personnel are identified as the high-risk group according to the Italian Ministry of Health guidelines. Additionally, our survey on vaccination adherence reveals that a higher VC could be reached by simply expanding access to Mpox vaccination to other HCWs.

We suggest that it could be very recommendable to extend the availability of smallpox vaccination to frontline HCWs, especially emergency medicine and infectious disease HCWs.

Further studies should be encouraged to explore the attitudes toward vaccination against Mpox in other Italian hospitals, to assess the motivation for compliance or not with the smallpox vaccination to protect against Mpox, and to compare it with the attitude towards other mandatory or recommended vaccinations for HCWs like, for example, COVID, flu, hepatitis B, and others.

## Figures and Tables

**Table 1 vaccines-11-01741-t001:** Differences in response rate by ward and job title.

	Answered	Not Answered	*p*	Odds Ratio	95% CI
Ward					
Infectious diseases	43 (82.7%)	9 (17.3%)		1	
Emergency medicine	116 (50%)	116 (50%)	<0.001	0.209	(0.098, 0.449)
Virology	38 (88.4%)	5 (11.6%)	0.44	1.591	(0.490, 5.162)
Proctological surgery	7 (63.6%)	4 (36.4%)	0.724	0.733	(0.130, 4.123)
Job title					
Physician	31 (46.3%)	36 (53.7%)		1	
Biologist	3 (100%)	0 (0%)	0.484	1.935	(0.304, 12.300)
Nurse	98 (59.8%)	66 (40.2%)	0.018	1.976	(1.123, 3.480)
Healthcare assistant	35 (55.6%)	28 (44.4%)	0.095	1.806	(0.901, 3.620)
Midwife	3 (100%)	0 (0%)	0.985	7.43 × 10^6^	(0, inf)
Laboratory technician	34 (94.4%)	2 (5.6%)	<0.001	43.871	(5.687, 338.450)

**Table 2 vaccines-11-01741-t002:** Distribution of vaccine and unvaccinated subjects in AOUP wards, in number percentage, *p*-value, odds ratio, and 95% confidence interval.

	Vaccinated	Not Vaccinated	*p*	Odds Ratio	95% CI
Age	54 (7)	37 (19)	<0.001	1.51	(1.330, 1.710)
Female	59 (40.4%)	87 (49.6%)		1	
Male	20 (34.5%)	38 (65.5%)	0.434	0.776	(0.412, 1.464)
Ward					
Infectious diseases	26 (60.5%)	17 (39.5%)		1	
Emergency medicine	38 (32.8%)	78 (67.2%)	0.002	0.319	(0.154, 0.657)
Virology	12 (31.6%)	26 (68.4%)	0.01	0.302	(0.121, 0.755)
Proctological surgery	3 (42.9%)	4 (57.1%)	0.388	0.49	(0.097, 2.470)
Job title					
Physician	12 (38.7%)	19 (61.3%)		1	
Biologist	0 (0%)	3 (100%)	0.986	2.75 × 10^−7^	(0, inf)
Nurse	32 (32.7%)	66 (67.3%)	0.536	0.768	(0.332, 1.770)
Healthcare assistant	22 (62.9%)	13 (37.1%)	0.052	2.679	(0.989, 7.260)
Midwife	1 (33.3%)	2 (66.7%)	0.855	0.792	(0.064, 9.710)
Laboratory technician	12 (35.3%)	22 (64.7%)	0.776	0.864	(0.315, 2.370)

**Table 3 vaccines-11-01741-t003:** Descriptive analysis results of the subgroups that want to get vaccinated and the subgroup that does not want to get vaccinated in the study population in number, percentage, *p*-value, odds ratio, and 95% confidence interval.

	Willingness to Vaccination	Denies Willingness	*p*	Odds Ratio	95% CI
Age	38 (17.5)	46 (18)	0.01	0.959	(0.930, 0.989)
Female	33 (22.6%)	113 (77.4%)		1	
Male	14 (24.1%)	44 (75.9%)	0.814	1.09	(0.533, 2.229)
Ward					
Infectious diseases	11 (25.6%)	32 (74.4%)		1	
Emergency medicine	25 (21.6%)	91 (78.4%)	0.59	0.799	(0.354, 1.807)
Virology	5 (13.2%)	33 (86.8%)	0.168	0.441	(0.138, 1.411)
Proctological surgery	6 (85.7%)	1 (14.3%)	0.012	17.455	(1.886, 161.528)
Job title					
Physician	11 (35.5%)	20 (64.5%)		1	
Biologist	1 (33.3%)	2 (66.7%)	0.941	0.909	(0.074, 11.194)
Nurse	21 (21.4%)	77 (78.6%)	0.118	0.496	(0.206, 1.195)
Healthcare assistant	8 (22.9%)	27 (77.1%)	0.261	0.539	(0.183, 1.584)
Midwife	3 (100%)	0 (0%)	0.985	1.05 × 10^7^	(0, inf)
Laboratory technician	3 (8.8%)	31 (91.2%)	0.015	0.176	(0.044, 0.710)

## Data Availability

Data is contained within the article.

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
