# Peer review of "Evaluation of Smallpox Vaccination Coverage and Attitude towards Monkeypox Vaccination among Healthcare Workers in an Italian University Hospital"

_vaccines, 2023, doi:10.3390/vaccines11121741_

Round 1
Reviewer 1 Report
Thank you for the opportunity to revise your manuscript. The topic is interesting, however, the writing and reporting must be improved.
The introduction is quite long. Please focus on the main aspects, strictly related to the topic of the manuscript.
Please, specify if the used questionnaire was a validated tool.
Please, add the full questionnaire as supplementary material.
Please, specify the administration way (on line, paper-based, ecc)
Please, specify the recruitment methods.
Please, specify if you collected data in anonymous way or not.
The statistical analyses provided are generally speaking correct, however to poor. Please, consider to perform a regression analysis.
Section 3.1 is out of the results section. Please remove, or if you consider it fundamental for you paper you can move the text in the methods and the table in supplementary materials.
Presentation of results is a bit confusing ,please revise
Multiple typos errors need to be corrected.
Lines 319-320 it is not clear what numbers in parentesis mean. Please revise
Discussion is very confusing and too many data, already shown in results section, are repited. Please, deeply revise this section. Be more succint and make an internal and external comparison of your results. In other words, try to understand consistency of your results simultaneously looking at your results and what is already known in literature.
In the discussion section a paragraph regarding implications for policies should be added in order to understand importance and impact of your results.
Conclusions should be shortened.
A general comment: the manuscript appears to be fragmentated. most of the sentences are not well connected with the previous, and each paragraph is frequently separated even when same concept is described. Please, revise the structure and the English.
Thank you for the opportunity to revise your manuscript. The topic is interesting, however, the writing and reporting must be improved.
Reviewer 2 Report
The study “Evaluation of Smallpox Vaccination Coverage and Attitude towards Monkeypox Vaccination among Healthcare Workers in an Italian University Hospital" critically examines vaccination adherence among healthcare workers in the context of Monkeypox (Mpox) infection risk within the AOUP hospital in Italy. It assesses vaccination rates and willingness among HCWs across different departments and job categories, exploring potential variations based on factors such as age and gender. However, there are certain shortcomings in the manuscript, which will enhance the clarity and importance of the manuscript.
The abstract mentions Smallpox vaccination in Italy without explaining why it's relevant in the context of Monkeypox. Since Smallpox has been eradicated, its mention here seems out of place and may confuse readers. The abstract briefly mentions the methodology as a self-report administered to healthcare workers (HCWs) but does not provide information about the design, sample size, or data collection process. The abstract presents data on vaccination coverage and attitudes in different departments but does not provide a clear and concise summary of the key findings. Data should be presented in a structured manner to facilitate understanding. The abstract concludes with a recommendation to extend Smallpox vaccination to frontline HCWs in Italy, but this recommendation lacks justification based on the presented data and findings. Additionally, the abstract contains some grammatical issues and awkward phrasing. It should be revised for clarity and readability.
The introduction discusses Smallpox extensively but does not explain the direct relevance of Smallpox vaccination to Monkeypox, which may confuse readers. The focus should be on Monkeypox, as this is the primary subject of the study. The introduction uses terms like "Mpox," "Monkeypox," and "Smallpox" interchangeably, which could create confusion. The terminology should be consistent throughout the text. The introduction mentions the study's aim to investigate Smallpox vaccination coverage and attitudes but does not clearly state-specific research questions or hypotheses. It should explicitly outline what the study intends to discover. The introduction presents information on vaccination strategies, laws, and guidelines in a somewhat disorganized manner. It would benefit from a more structured presentation to improve clarity. Furthermore, the authors have not sufficiently explained why this study is important or what knowledge gaps it intends to fill. What is the significance of assessing vaccination coverage and attitudes in Italian healthcare workers in the context of Monkeypox?
The methodology describes data collection in general terms but lacks specific details about how the questionnaire was developed, the content of the questions, and the validation process (if any). Providing more information on the questionnaire's design and content would enhance the transparency of the study. The methodology mentions that healthcare workers (HCWs) from departments at both high and low/moderate risk for Monkeypox exposure were considered, but it doesn't explain how these departments were selected or why they were chosen. Providing a clear rationale for the department selection process is important. The methodology mentions a self-declaration form distributed to HCWs, but it does not provide information on the response rate or efforts made to maximize participation. The response rate can affect the study's representativeness and validity. There is no mention of ethical approval or institutional review board (IRB) clearance for the study. Research involving human subjects typically requires ethical review and approval, and this should be addressed in the methodology. The methodology does not explain how the sample size was determined. A justification for the chosen sample size, based on power calculations or statistical considerations, would strengthen the study's validity.
The results mention that 204 out of 338 healthcare workers (HCWs) provided the requested data through self-certification. While this response rate is mentioned, there is no discussion of potential biases introduced by nonresponse. It's essential to address whether the respondents are representative of the entire HCW population in the study, as nonresponse bias can affect the validity of the findings. The results indicate a significant age difference between vaccinated and unvaccinated HCWs. However, the implications of this age difference are not explored in the results section. Discussing potential reasons for this age disparity and its relevance to Monkeypox risk would enhance the interpretation of results. Furthermore, is suggested that in Tables 2, 3, 4, and 5, Adjusted Odd ratios regarding the observed variable. Adding a separate column regarding the dichotomous and polychotomous variables would increase the clarity of the results.
The discussion section of the manuscript provides a thorough analysis of the results obtained in the study, discussing various aspects related to vaccination coverage (VC) and willingness to vaccinate against Mpox among healthcare workers (HCWs) at AOUP. However, there are also some shortcomings and areas that could be further improved:
The study is based on data from a single hospital (AOUP) in Italy. While the findings are interesting and relevant for this specific hospital, it is essential to acknowledge that the results may not be generalizable to other hospitals or regions. The discussion should explicitly state this limitation and emphasize the need for multicenter studies to validate the findings and provide a broader perspective. The discussion mentions that only 60.36% of HCWs completed the self-declaration. This response rate raises concerns about potential non-responders. The discussion should elaborate on how this response rate might affect the study's validity and the interpretation of results. The discussion highlights the differences in VC and willingness to vaccinate among various departments and job categories but does not delve deeply into the reasons behind these differences. Understanding the motivations and barriers to vaccination among different groups of HCWs could provide valuable insights for developing targeted interventions to improve vaccine coverage. This aspect should be discussed or suggested for future research. The discussion mentions that the study did not evaluate HCWs' knowledge and perceptions regarding the Mpox virus. Including an assessment of HCWs' understanding of the disease and their perceptions of the vaccine's safety and efficacy would provide a more comprehensive picture of the factors influencing vaccination decisions. This information could be valuable for future research or interventions. While the discussion acknowledges certain limitations, it could benefit from explicitly outlining recommendations for future research. This could include suggestions for larger, multicenter studies, qualitative research to explore reasons behind vaccination decisions, and efforts to assess HCWs' knowledge and perceptions regarding Mpox.
The conclusion suggests extending Smallpox vaccination access to all frontline HCWs involved in patient care based on the ECDC guidelines. While this is a valuable recommendation, it might benefit from additional contextualization and justification. The discussion could elaborate on how this policy change aligns with best practices in other countries or similar healthcare settings. Providing evidence or examples of successful policy changes in other regions could strengthen the argument for this proposal.
Similar to the discussion section, the conclusion should explicitly acknowledge the limitation of generalizability. While the study's findings and recommendations are relevant to AOUP, they may not apply universally to all healthcare settings in Italy or other countries. Emphasizing the need for further research and adaptability to different contexts is important. The conclusion could benefit from a section dedicated to outlining specific avenues for future research. What are the unresolved questions or areas that require further investigation? Suggesting potential research questions or study designs could guide researchers and policymakers in advancing this field.
Moderate editing in the English language is required.
Reviewer 3 Report
The manuscript (ID: vaccines-2611935) aimed to assess smallpox vaccination coverage among health care workers in an Italian hospital with different levels of risk of exposure to Mpox infection and to assess their attitudes toward vaccination.
In this paper, except for the Introduction section, all other parts of this paper (Abstract, Methods, Results, Discussion, and Conclusions) require extensive revision.
Comments (Major revision):
- Lines 17-20: Specify the study design used in this paper.
- Section Introduction: The paper provides a comprehensive overview of the occurrence of the multi-country outbreak of Mpox infection in 2022, the clinical characteristics of Mpox, the epidemiology of Mpox, as well as the possibilities of prevention and control of Mpox. A special subsection in the Introduction section refers to the presentation of Mpox in Italy. At the end of the Introduction section, the goal of this paper is clearly defined.
- Line 81: State that it is the year 2022 when the European Medicines Agency made the above decision.
- Line 155: Mandatory:
¾ Specify a new subsection `Study population`.
¾ List the inclusion and exclusion criteria in this study.
¾ Specify new subsection `Study design`.
¾ List the `Study sample` subsection with appropriate characteristics.
¾ Specify `Participation rate` and `Response rate`.
- Lines 162-164: Must state the definition and criteria you applied when you considered a person (HCW) vaccinated against smallpox, citing the appropriate reference.
- Lines 204-219: Explain the importance, justification and need of the data presented on Table 2 in the context of what was stated on Lines 168-171.
- Table 2: Check and correct values for variable `Infectious diseases`.
- Lines 274-377: Unnecessary and pointless repetition and description of own results, which have already been shown on the previous pages of this work. Taken as a whole, the Discussion section did not satisfactorily present the comparison of its own results with the results of similar studies in the world. Also, the complete lack of possible explanations for the differences in the results of this and similar studies is evident.
- Lines 378-387: Discuss sources of bias (such as information bias, non-response bias, etc.) as Limitations in this study.
- Lines 388-411: Overly extensive Conclusions section. Unnecessary and excessive repetition of the previously mentioned results. Highlight the most important results.
The quality of English language is appropriate.
Round 2
Reviewer 1 Report
The manuscript has been improved, I am satisfied with the reviews provieded
Reviewer 2 Report
The authors have made significant improvements to the manuscript since its initial submission. They have diligently addressed the comments and suggestions, resulting in a substantially enhanced version of the paper. The clarity and organization of the content have notably improved, making the manuscript more accessible and engaging for readers. I am pleased to recommend the manuscript for publication in its present form.
Reviewer 3 Report
Thank you for the opportunity to review the revised version of this manuscript.
There are no significant improvements made in the paper. The Authors have not addressed my comments in a satisfactory manner, nor have they provided a satisfactory explanation. In the present form, the quality of the presented results is of meaning for a journal of local character. For the important topic it covers, the paper does not provide information that is for publishing as an original paper in a renowned journal such as Vaccines.
The quality of English language can not be assesed due to anappropriate use of track changes.